# Object-Centric Neuro-Argumentative Learning

**Abdul Rahman Jacob**                    ABDUL-RAHMAN.JACOB20@IMPERIAL.AC.UK
*Imperial, London, UK*

**Avinash Kori**                                    A.KORI21@IC.AC.UK
*Imperial, London, UK*

**Emanuele De Angelis**                  EMANUELE.DEANGELIS@IASI.CNR.IT
*IASI-CNR, Rome, Italy*

**Ben Glocker**                                  B.GLOCKER@IMPERIAL.AC.UK
*Imperial, London, UK*

**Maurizio Proietti**                      MAURIZIO.PROIETTI@IASI.CNR.IT
*IASI-CNR, Rome, Italy*

**Francesca Toni**                                      FT@IC.AC.UK
*Imperial, London, UK*

**Editors:** Leilani H. Gilpin, Eleonora Giunchiglia, Pascal Hitzler, and Emile van Krieken

## Abstract

Over the last decade, as we rely more on deep learning technologies to make critical decisions, concerns regarding their safety, reliability and interpretability have emerged. We introduce a novel Neural Argumentative Learning (NAL) architecture that integrates Assumption-Based Argumentation (ABA) with Object-Centric (OC) deep learning for image analysis. Our *OC-NAL* architecture consists of neural and symbolic components. The former segments and encodes images into facts, while the latter applies ABA learning to develop ABA frameworks enabling image classification. Experiments on synthetic data show that the OC-NAL architecture can be competitive with a state-of-the-art alternative. The code can be found at https://github.com/AbdulRJacob/Neuro-AL

## 1. Introduction

Over the last decade, AI, supported by deep learning, has become increasingly more prevalent in our lives. However, as we rely more on deep learning technologies to make critical decisions, concerns regarding their safety, reliability and explainability naturally emerge. Indeed, deep learning models, such as those used for image classification, are considered black boxes as their internal workings are not easily interpretable, resulting in a possible lack of trust in their predictions.

Motivated by the need for more interpretable image classifiers, we introduce a novel neuro-argumentative learning (NAL) architecture which generates symbolic representations in the form of assumption-based argumentation (ABA) frameworks (Dung et al., 2009) from images, using objects identified in these images by Object-Centric (OC) methods (De Vita, 2020). The resulting ABA frameworks can be used to make predictions while allowing humans to follow a line of reasoning as to why the model made those predictions.

To generate ABA frameworks, our *OC-NAL* architecture uses ABA Learning (De Angelis et al., 2023, 2024), a method that uses argumentation in a logic-based learning fashion

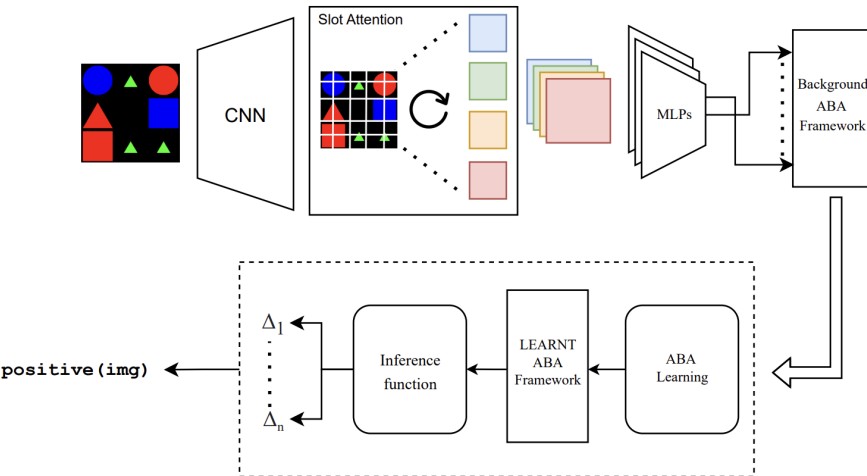

Figure 1: OC-NAL: the input image (top left) is processed by slot attention to obtain objects (coloured squares) mapped by MLPs into facts fed into a Background ABA framework; then ABA learning generates a Learnt ABA framework, which may admit several 'extensions' (i.e., sets of accepted arguments) $\Delta_1, \ldots, \Delta_n$; inference therewith gives a classification (bottom left).

to generate ABA frameworks which, with their accepted arguments, cover given positive examples and do not cover given negative examples. OC-NAL also uses slot attention (Locatello et al., 2020) as the underpinning OC method, to support a granular understanding of input images in terms of the objects they contain. Overall, our OC-NAL architecture enables the extraction of meaningful properties and relationships between objects within the images, facilitating accurate classification with interpretable argumentation frameworks.

**Contributions** Overall, we make the following contributions: 1) we tailor slot-attention to generate factual background knowledge suitable to be injected in ABA Learning; 2) we combine slot-attention and ABA Learning into a pipeline architecture (that we term *OC-NAL*) for neuro-argumentative learning; 3) we assess the performances of our method experimentally on (synthetic) image datasets, showing that it can be competitive against a baseline (Neuro-Symbolic Concept Learner (NS-CL) (Mao et al., 2019)).

## 2. Related Work

Proietti and Toni (2023) overview several approaches integrating logic-based learning with image classification. Specifically, NeSyFold (Padalkar et al., 2024) uses the rule-based learning algorithm FOLD-SE-M to convert binarised kernels from a trained CNN to a set of ASP rules (Gelfond and Lifschitz, 1991) with abstract predicates. It then uses semantic labelling to assign human-like concepts to predicates leading to global explanations of image classifications. Further, Embed2Sym (Aspis et al., 2022) uses clustered embeddings extracted by a neural network in combination with a symbolic reasoner encoded with predefined rules for explainable predictions. Some other approaches integrate learning and argumentation for

image classification, notably Thauvin et al. (2024) classify images based on argumentative debates drawn from encoders and Kori et al. (2024a) explain the outputs of image classifiers with argumentative debates drawn from quantized features. *No existing approach uses argumentation with object-centric methods, as we do.* Indeed, to the best of our knowledge we are the first to propose such combination as a neuro-symbolic learning approach.

The closest approach to OC-NAL is Neuro-Symbolic Concept Learner (NS-CL) (Mao et al., 2019), which we use as a baseline. NS-CL uses object-centric learning via slot attention to identify objects within images, while its reasoning modules incorporate set transformers to extract and generate explanations. These are presented as grid-like representations, enabling users to understand the concepts applied in classification and derive further rules. We use this approach as a baseline for the evaluation of our method, even though its outputs are at a different level of abstraction than our fully argumentative method.

## 3. Background

In this section we briefly recall some notions that are the basis of our OC-NAL architecture.

**Slot Attention** Slot Attention (Locatello et al., 2020) maps a set of $N$ input feature vectors $z \in \mathbb{R}^{N \times d}$, of dimension $d$, obtained from an input image $x$, to a set of $K$ output vectors, of dimension $d_S(\leq d)$, $\hat{z} \in \mathbb{R}^{K \times d_S}$ that we refer to as slots. The input features are projected with linear layers to create *key* and *value* vectors, represented by $\mathbf{k}$ and $\mathbf{v}$, respectively. The slots are also projected with a linear layer, resulting in a *query* vector $\mathbf{q}$. To simplify our exposition, later on, let $f_s$ denote the *slot update* function, defined as:

$$\hat{z}^{t+1} := f_s(z, \hat{z}^t) = \hat{A}\mathbf{v}, \ \hat{A}_{ij} := \frac{A_{ij}}{\sum_{l=1}^{N} A_{il}}, \ A := \text{softmax}\left(\frac{\mathbf{q}\mathbf{k}^T}{\sqrt{d_S}}\right) \tag{1}$$

where $A \in \mathbb{R}^{K \times N}$ is the cross-attention matrix. The queries $\mathbf{q}$ in slot attention are a function of the slots $\hat{z}^t$, and are iteratively refined over $T$ iterations. The initial slots $\hat{z}^{t=0}$ are randomly sampled from a standard Gaussian (Locatello et al., 2020).

**Assumption-Based Argumentation (ABA)** Assumption-Based Argumentation (Dung et al., 2009) is a well-known symbolic formalism for modelling non-monotonic reasoning. An ABA framework (Dung et al., 2009) is a tuple $\langle \mathcal{L}, \mathcal{R}, \mathcal{A}, \overline{\phantom{\ }} \rangle$ where: (i) $\langle \mathcal{L}, \mathcal{R} \rangle$ is a deductive system where $\mathcal{L}$ is the language and $\mathcal{R}$ is the set of (inference) rules; (ii) $\mathcal{A} \subseteq \mathcal{L}$ is the set of assumptions; (iii) $\overline{\phantom{\ }}$ is a total mapping from $\mathcal{A}$ to $\mathcal{L}$ where $\bar{a}$ is referred to a contrary of $a$ ($a \in \mathcal{A}$). In this paper, we consider *flat* ABA frameworks, where assumptions are not heads of rules. Also, we assume that the elements of $\mathcal{L}$ are *atoms* and, for the sake of simplicity, we omit the language as it can be derived from $\langle \mathcal{R}, \mathcal{A}, \overline{\phantom{\ }} \rangle$. We illustrate with a simple example, where, as in (Dung et al., 2009), we use schemata to write rules, assumptions and contraries, using the variable A.

**Example 1** *A simple ABA framework for image classification is $\langle \mathcal{R}, \mathcal{A}, \overline{\phantom{\ }} \rangle$, where:*

$\mathcal{R} = \{ \quad \rho_1 : circle(A) \ :- \ A=img\_1, \qquad \rho_2 : circle(A) \ :- \ A= img\_2,$
$\qquad \quad \rho_3 : square(A) \ :- \ A= img\_2, \qquad \rho_4 : c\_1(A) \ :- \ circle(A), \ alpha(A),$
$\qquad \quad \rho_5 : c\_alpha(A) \ :- \ square(A) \quad \}$

$\mathcal{A} = \{ alpha(A) \} \qquad \overline{alpha(A)} = c\_alpha(A)$

Here, $\mathcal{R}$ is a set of *rules*, *each with a name* $\rho_i$, *a head following* `:` *and a body following* `:-`, $\mathcal{A}$ *is a set of* assumptions, *in this case consisting of a single assumption, with each assumption equipped with a* contrary, *in this case the contrary of* `alpha(X)` *is* $\overline{\texttt{alpha(X)}}$. *The intuirive meaning of the ABA framework is as follows: images* `img_1` *and* `img_2` *contain a circle* ($\rho_1$, $\rho_2$), *image* `img_2` *contains a square* ($\rho_3$), *and image* `A` *belongs to concept* `c_1`, *if it contains a circle, unless it also contains a square* ($\rho_4$, $\rho_5$).

We define a *fact* as a rule with distinct variables in the head and only equalities in the body.

To decide which conclusions may be drawn from an ABA framework, arguments and attacks between them are first obtained, then acceptance of arguments is determined using an extension-based semantics, in our case of *stable extensions* (Dung et al., 2009). An *argument* for the claim $c \in \mathcal{L}$ supported by $A \subseteq \mathcal{A}$ and $R \subseteq \mathcal{R}$ (denoted as $A \vdash_R c$) is a finite tree with nodes labelled by sentences in $\mathcal{L}$ or by $\tau$ denoting *true*, the roots labelled by $c$, the leaves either *true* or assumptions in $A$, and non-leaves $c'$ with, as children, the elements of the body of some rule in $R$ with the head $c'$. An argument $A \vdash_R c$ *attacks* an argument $A' \vdash_{R'} c'$ iff there is an assumption $a \in A'$ such that $\bar{a} = c$. A set of arguments $E$ is *stable* iff the set is conflict-free (i.e. no argument in $E$ attacks an an argument also in $E$) and for every argument not in $E$ there is an argument in $E$ attacking it. We illustrate these notions with the earlier example.

**Example 2** *The following arguments can be obtained (amongst others) from the ABA framework in example 1:*

$\{\texttt{alpha(img\_1)}\} \vdash_{\{\rho_1,\rho_4\}} \texttt{c\_1(img\_1)}, \quad \{\texttt{alpha(img\_2)}\} \vdash_{\{\rho_2,\rho_4\}} \texttt{c\_1(img\_2)},$
$\{\} \vdash_{\{\rho_2\}} \texttt{circle(img\_2)}, \qquad\qquad\quad \{\} \vdash_{\{\rho_3,\rho_5\}} \texttt{c\_alpha(img\_2)}.$

*Intuitively, each argument is a deduction from (possibly empty) sets of assumptions (the premises) to claims (e.g.* `c_1(img_1)` *for the first argument), using sets of rules. Attacks between arguments result from undercutting assumptions in the premises of arguments. Here, the fourth argument attacks the second, as the former is a deduction of the contrary of the assumption occurring in the premise of the latter. The third and fourth arguments belong to the single stable extension admitted by this simple ABA framework, as they cannot be attacked by any other arguments.*

**Learning ABA Frameworks** We use the ASP-ABALearn method by De Angelis et al. (2023, 2024). This takes in input a Background ABA framework (admitting at least one stable extension), sets $\mathcal{E}^+$ and $\mathcal{E}^-$ of positive and negative examples (i.e., atoms obtained from labelled images), respectively, and returns in output a Learnt ABA framework (admitting at least one stable extension) such that all positive examples are accepted in all stable extensions and no negative example is accepted in all the stable extensions. Computationally, ASP-ABALearn leverages the fact that flat ABA frameworks (where assumptions cannot be claims of arguments supported by other assumptions) can be mapped to logic programs. This is done by replacing each assumption $\alpha(X)$ with *not* $p(X)$ where $\overline{\alpha(X)} = p(X)$.

## 4. OC-NAL Architecture

We will now explain the OC-NAL architecture shown in Figure 1, by detailing the neural and symbolic components and their training, as well as inference post-learning.

**Inputs** The OC-NAL architecture accepts a dataset $D \subseteq X \times Y \times L$ of labelled images, where $X$ is the given set of images, $L = \{c_1, c_2\}$ is a set of classes, and $Y = \{0, 1\}^{K \times (P+1)}$, for $K$ the total number of objects/slots that may occur in images, and $P$ the total number of *properties* that each of these objects may have (we consider an extra property for characterising the absence of objects). As a simple example, for images such as the one in Figure 1, $K = 10$ (as there are a maximum of 9 objects in each such image plus the background) and $P = 8$ (3 for the shapes, 3 for the colours, 2 for sizes). We assume that $D$ is not noisy. $L$ is a set of two alternative classes. $Y$ is a metadata consisting of one-hot encodings, each representing all objects and their corresponding properties in an image. Note that the neural component of the architecture disregards the labels in $L$, focusing instead on the images in $X$ in a weakly supervised manner, while the symbolic component disregards the image itself, using instead its abstraction drawn from the neural component.

**Neural Component** This uses a Convolutional Neural Network (CNN), slot attention and a set of multi-layer perceptrons (MLP) to convert a given input image $x$ into facts for the symbolic component, amounting specifically to the Background Knowledge for ABA Learning or, during inference, facts to be added to the generated ABA framework.

First, the input image is converted into features $z$ using a CNN, which is further used by the slot attention model trained using the process described in Section 3 to produce slots $\hat{z}$.

Then, each slot $\hat{z}$ is passed through the MLPs to extract the properties of each object. To identify both continuous and categorical properties, we use MLPs of two types: classification MLPs, which use a softmax activation in the final layer to predict the most likely attribute for a given slot and regression MLPs, to predict the location of objects and determine whether each given slot has attended to a real object in the image. The results of these predictions are then concatenated to form the final prediction $\hat{y}$ for the input image, with the corresponding ground truth $y \in Y$. This component is trained with weak-supervision and is optimised by minimising the loss function:

$$\text{MSE}(x, \hat{x}) + \alpha \min_{\tau \in S_K} \sum_{j=1}^{P} \text{BCE}(y_j, \tau(\hat{y})_j) \tag{2}$$

which encapsulates both training objectives. The first is the mean square error (MSE) between the input image $x$ and the (reconstructed) image $\hat{x}$, ensuring the reconstruction quality (from the slots) of the model. The second is the binary cross entropy (BCE) between the ground truth label $y$ and the predicted label $\hat{y}$, where $y_j$ corresponds to a particular property in vector $y$ and $\tau(\hat{y})$ is the prediction for a permutation of the $K$ objects, drawn from $S_K$, which denotes the set of all such permutations of $K$ objects. Intuitively, we compare each of the permutations with the ground-truth representation (with a specific order). Given the equivariance property of slot attention, we need to first align the slots before applying the BCE loss. To circumvent this, we used the Hungarian matching algorithm (Kuhn, 2010). Finally we balance both loss terms with the hyperparameter $\alpha$.

**Symbolic Component** This receives the output predictions from the neural component and transforms them into facts for the Background ABA Framework taken in input by the ASP-ABALearn algorithm. It also uses the input labels in $L$ to obtain appropriate positive and negative examples in $(\mathcal{E}^+, \mathcal{E}^-)$. This is accomplished by aggregating the slots and performing K-means clustering. The number of clusters corresponds to the desired

number of examples, and an image is chosen from each cluster as a representative positive or negative example. We also check the confidence of each prediction and prune off any image below a certain threshold.

The slot predictions are then passed to concept embedding functions which use a dictionary (for the $K$ objects and the $P$ properties) to convert the raw predictions to ABA facts. For each image and object, an identifier is given in the form of an atom `image(img_i)` and a constant `object_i` respectively. Then, for each slot prediction, we take the argmax to identify the properties in the dictionary which are attributed to the object. We encode this as a fact, e.g. `blue(object_i)`.

Once all images and objects are encoded into the Background ABA Framework, we generate the ASP-ABALearn command `aba_asp('filename.aba', e_pos,e_neg)`. This specifies which images are positive/negative, using `e_pos`/`e_neg` as the encodings of $\mathcal{E}^+/\mathcal{E}^-$, as discussed earlier. The ABA-ASPLearn algorithm is then run to produce a Learnt ABA Framework.

**Inference** At inference time, we run a slightly different pipeline to obtain a final classification for each unseen input image. Specifically:

1. We pass the image through the neural component to obtain predictions of the objects and their properties therein. These are subsequently converted into facts as during training.

2. We then create an ABA Framework which contains these facts, the rules learnt via ASP-ABALearn and any extra background knowledge.

3. The stable extensions of this ABA framework are then computed (using a straightforward mapping into ASP, and then using an ASP solver. specifically we use Clingo (Gebser et al., 2019)).

4. Depending on the ABA framework, we may obtain more than one stable extension. The prediction boils down to checking whether the atom sanctioning that the input image belongs to concept $c_1$ is a member of all the stable extensions (i.e., it is a *cautious* consequence of the Learnt ABA framework).

## 5. Experimental Evaluation

We conducted experiments on the OC-NAL architecture to answer the following questions:

**Q1**: How well can the Neural Component identify/predict object properties?

**Q2**: How well can the OC-NAL architecture learn ABA frameworks which describe the latent rules in images so that they be used for classification?

**Q3**: How well does our OC-NAL architecture scale w.r.t. the number of examples used in the Symbolic Component and the complexity of the latent rules?

| SHAPES Dataset ASP Rules |
|---|
| `s1(A) :- image(A), in(A,B), square(B), blue(B).` |
| `s2(A) :- image(A), in(A,B), triangle(B), small(B), green(B).` |
| `s3(A) :- image(A), in(A,B), in(A,C), triangle(B), blue(B), circle(C), red(C), large(C).` |
| `s4(A) :- image(A), in(A,B), in(A,C), circle(B), red(B), square(C), blue(C), above(B,C).` |
| `s5(A) :- image(A), in(A,B), in(A,C), triangle(B), red(B), circle(C), green(C), left(B,C).` |
| `s6(A) :- not exception(A), image(A).` |
| `exception(A) :- image(A), in(A,B), circle(B), blue(B).` |

Figure 2: ASP rules used to generate the 6 classes for the SHAPES dataset. We generated 3K images for each rule half of which was the negative instance of the rules. We then took 500 positive and negative splits for each rule as testing data. In total, we had 12K images for training and 6K for testing.

**Experiments** To address these questions, we defined various binary classification tasks using our adaptation of the SHAPES dataset (Andreas, 2017) . This dataset was generated by a tool that processed ASP rules to create images conforming to them. We defined 6 rules, as detailed in Figure 2, to form the dataset. Each binary classification task aimed to distinguish between positive and negative instances for each class. This dataset served as a baseline to evaluate the viability of using argumentation to reason in an object-centric way. We also defined a multi-class classification task on the CLEVR dataset using CLEVR-Hans3 (Stammer et al., 2021) which splits CLEVR images into 3 classes based on the following concepts $c_1$: `Large (Gray) Cube and Large Cylinder`, $c_2$: `Small metal Cube and Small (metal) Sphere`, $c_3$: `Large blue Sphere and Small yellow Sphere`. The goal of this task was for OC-NAL to generate Learnt ABA frameworks that could differentiate these classes.

**Setup** We trained the OC-NAL architecture in two stages as described in Section 4. The Neural Component was trained using the full datasets for 1000 epochs with hyperparameter $\alpha = 0.35$. The Symbolic Component used 10 positive and 10 negative examples from the datasets to obtain the Learnt ABA framework. Regarding the CLEVR-Hans3 task (which was multi-classed), we executed the Symbolic Component twice, the first run to distinguish c3 images from c1 and c2 and the second run to distinguish between c1 and c2, thus producing two frameworks. We compare the results with a ResNet (He et al., 2016) and the NS-CL.

**[Q1] Object Prediction** Figure 3 shows that the Neural Component effectively segmented images into their constituent objects and accurately predicted each object's property. We evaluated its localisation and segmentation capabilities using the Adjusted Rand Index, which measures similarity between clusters (with clusters representing objects and data points representing pixels), and the Average Precision metric. The scores ranged from 0.80 to 0.95, indicating consistent performance regardless of the number of objects present in each dataset.

We also observed high scores in standard metrics, indicating that the MLPs accurately predicted each object's properties. The F1 score exceeded 70%, suggesting that the model effectively minimises false positives and negatives, ensuring the symbolic component contains facts that accurately represent the image. Results on the CLEVR dataset were slightly

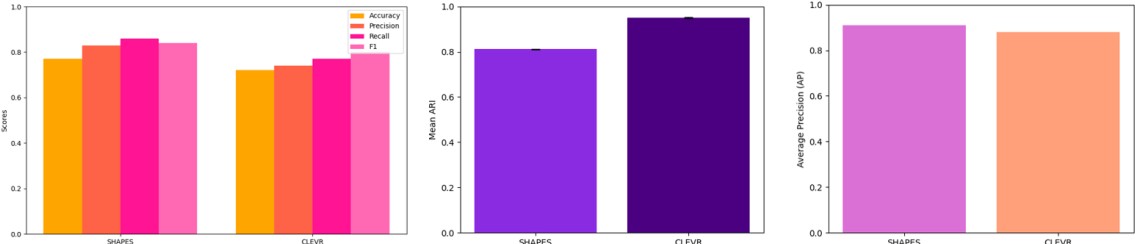

Figure 3: **(Left)** Standard machine learning evaluation metrics of attribute classification on datasets SHAPES and CLEVR **(Center)** Mean Adjusted Rand Index on datasets **(Right)** Average Precision on datasets.

worse than those on the SHAPES dataset, likely due to CLEVR objects having a larger number of properties. This suggests that the number of properties can affect prediction performance.

**[Q2] Classification**    From Table 1 we observe that the Learnt ABA frameworks performed well on the binary classification tasks for SHAPES, achieving near-perfect scores on most tasks. Among the metrics, we saw that recall was consistently the lowest. This is likely due to errors propagated from the Neural Component, which resulted in facts that did not accurately represent the images, causing some instances to be misclassified.

However, we saw the opposite when predicting classes s4 and s5, with significantly lower precision. We believe that this is due to the fact that in those tasks ASP-ABALearn learnt rules capturing (some of) the exceptions rather than the full concepts. For example, s1 images were defined in the dataset (see Figure 2) as those with blue squares. However, the learnt framework defined `s_1` as images with squares that are not red and not green (see `c_alpha2` in Figure 4). Despite being semantically equivalent, this reasoning made interpretation difficult and impacted the generation of more complex rules. Consequently, the F1 scores for these tasks dropped significantly due to the inability to capture all rule exceptions, leading to many false positives and thus lower precision.

We encountered a similar outcome with the learnt ABA frameworks generated for the CLEVR-Han3 tasks. These only partially captured the rules defining each class. For instance, the framework for c1 stated that these images contain cubes that are not small. However, this rule failed to distinguish some c2 images, which could also contain cubes that are not small. As a result, many c2 images were incorrectly classified as c1, as shown in the confusion matrix (see Figure 7).

Despite this issue, the learnt ABA frameworks were able to capture most of the facts for classifying c3 images, leading to an F1 score of 0.68 (see Table 2). This outperformed the ResNet baseline, though it was worse than NS-CL, which scored above 0.80%. The superior performance of NS-CL could be attributed to its use of a set transformer for classification, rather than relying solely on symbolic reasoning.

**[Q3] Scalability**    During our experiment, we found that the symbolic component, specifically ASP-ABALearn, faced some scalability issues as the execution time grew significantly as we increased the number of examples. This could be due to a larger search space gen-

```
% Learnt Rules
s_1(A) :- in(A,B), square(B), alpha_2(B,A).
c_alpha_2(A,B) :- image(B), red(A).
c_alpha_2(A,B) :- image(B), green(A).
```

Figure 4: Rules in the ABA framework generated by our OC-NAL architecture for SHAPES (class s1). Here, `alpha_2` is an assumption, with $\overline{\texttt{alpha\_2(A,B)}}$=`c_alpha_2(A,B)`.

```
% Learnt Rules
c_1(A) :- in(A,B), cube(B), alpha_2(B,A).
c_alpha_2(A,B) :- small(A), image(B).
```

Figure 5: Rules in the ABA framework generated by our OC-NAL architecture for SHAPES (class s5). Here, `alpha_2` is an assumption, with $\overline{\texttt{alpha\_2(A,B)}}$=`c_alpha_2(A,B)`.

```
% Learnt Rules
c_3(A) :- in(A,B), sphere(B), alpha_2(B,A).
c_alpha_2(A,B) :- brown(A), image(B).
c_alpha_2(A,B) :- green(A), image(B).
c_alpha_2(A,B) :- cyan(A), image(B).
c_alpha_2(A,B) :- red(A), image(B).
c_alpha_2(A,B) :- large(A), image(B).
c_alpha_2(A,B) :- blue(A), image(B).
c_alpha_2(A,B) :- gray(A), image(B).
```

Figure 6: Rules in the ABA framework generated by our OC-NAL architecture for CLEVR, differentiating class c3 from c1 and c2. Here, `alpha_2` is an assumption, with $\overline{\texttt{alpha\_2(A,B)}}$=`c_alpha_2(A,B)`.

| Predicted class | Accuracy | Precision | Recall | F1-Score |
|---|---|---|---|---|
| s1 | $99.0 \pm 0.0$ | $100.0 \pm 0.0$ | $97.5 \pm 0.0$ | $99.0 \pm 0.2$ |
| s2 | $96.0 \pm 0.0$ | $99.0 \pm 0.0$ | $93.5 \pm 0.0$ | $96.0 \pm 0.2$ |
| s3 | $98.0 \pm 0.0$ | $100.0 \pm 0.0$ | $97.5 \pm 0.2$ | $98.0 \pm 0.0$ |
| s4 | $75.0 \pm 3.0$ | $61.0 \pm 5.2$ | $98.5 \pm 0.4$ | $75.0 \pm 0.2$ |
| s5 | $86.0 \pm 4.1$ | $77.0 \pm 3.8$ | $96.5 \pm 0.2$ | $84.0 \pm 0.2$ |
| s6 | $99.0 \pm 0.0$ | $98.0 \pm 0.0$ | $100.0 \pm 0.0$ | $99.0 \pm 0.0$ |

Table 1: Standard evaluation metrics denoting how well OC-NAL can distinguish between positive and negative instances of rules present in SHAPES images

|          | Accuracy     | Precision    | Recall       | F1-Score     |
|----------|--------------|--------------|--------------|--------------|
| OC-NAL   | $69.1 \pm 0.3$ | $70.0 \pm 0.1$ | $69.1 \pm 0.2$ | $68.0 \pm 0.2$ |
| ResNet   | $65.2 \pm 0.3$ | $66.2 \pm 0.4$ | $65.2 \pm 0.2$ | $61.0 \pm 0.2$ |
| NS-CL    | $84.7 \pm 0.1$ | $86.1 \pm 0.2$ | $84.7 \pm 0.2$ | $84.0 \pm 0.2$ |

Table 2: Standard evaluation metrics for the performance of OC-NAL, ResNet, and NS-CL on the CLEVR-Hans3 task.

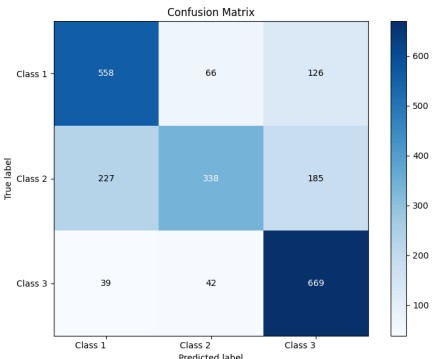

Figure 7: Confusion Matrix of OC-NAL on the CLEVR-Hans3 task.

erated when looking for ABA frameworks whose extensions cover all positive examples and none of the negative examples. It may happen that this search is unsuccessful and ASP-ABALearn halts with failure.

Our experiments also showed that the non-determinism present when generalising rules led to variability in both the quality of the results and the system's execution time. This variability was amplified by the size of the Background ABA framework. Additionally, we observed that longer execution task times negatively impacted the quality of the results, as heavily nested rules caused the frameworks to overfit to the example set.

## 6. Conclusions and Future Work

We have proposed a novel neuro-argumentative learning (OC-NAL) method for image classification, in the spirit of (Proietti and Toni, 2023), integrating slot attention for object identification in images by De Vita (2020), and the ASP-ABALearn implementation of ABA Learning by De Angelis et al. (2023, 2024). The proposed framework follows a faithful and human-understandable reasoning process. We empirically demonstrate that our approach can be effective via experiments with the resulting architecture on datasets of synthetic images.

Overall, we believe that this work has contributed some insights into the potential of argumentation in the space of image classification, demonstrating that an object-centric approach combined with ABA is a viable approach for neuro-symbolic learning. At the same time, several avenues for future work remain open. Specifically, we plan to extend our framework to deal with real images, rather than synthetic images. Also, we plan to explore novel instances of our general architecture by considering other forms of slot-attention, e.g. the method of (Kori et al., 2024b), and bespoke forms of ABA Learning suited to our setting. Furthermore, it would be interesting to explore variants of our approach where slot-attention and ABA learning are trained together, in an end-to-end fashion. Finally, we plan to explore the explainability of classification with our form of NAL, especially in comparison with the NS-CL baseline (Mao et al., 2019)).

## Acknowledgements

We thank support from the Royal Society, UK (IEC\R2\222045 - International Exchanges 2022). Toni was partially funded by the ERC under the EU's Horizon 2020 research and innovation programme (grant agreement No. 101020934) and by J.P. Morgan and the Royal Academy of Engineering, UK, under the Research Chairs and Senior Research Fellowships scheme. Kori was supported by UKRI through the CDT in Safe and Trusted Artificial Intelligence. This paper has also been partially supported by the Italian MUR PRIN 2022 Project DOMAIN (2022TSYYKJ, CUP B53D23013220006, PNRR M4.C2.1.1) funded by the European Union – NextGenerationEU, and by the PNRR MUR project PE0000013-FAIR - Future Artificial Intelligence Research (CUP B53C22003630006). De Angelis and Proietti are members of the INdAM-GNCS research group.

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
