# OpenReview forum: "Object-Centric Neuro-Argumentative Learning"
_nesyconf.org/NeSy/2025/Conference — NeSy 2025 Poster_

### Official Review · Reviewer_ZBBV · 2025-03-31
**Argumentation-based Neurosymbolic Approach Limited by Scalability, Generalization and Comparison**

**Rating:** 6
**Confidence:** 4

**Review:**

Summary:
The paper introduces a novel Neural Argumentative Learning (NAL) architecture combining object-centric learning with Assumption-Based Argumentation (ABA). While conceptually appealing, experimental limitations issues with scalability and generalization are major impediments in its usefulness. Furthermore the work lacks a comparison to neurosymbolic Inductive Logic Programming (ILP) methods, which are very similar to the work presented.

Review:
The authors propose an integration of deep learning and argumentation frameworks for enhanced interpretability and explainability in image classification tasks. The approach is novel in its use of ABA learning, bringing argumentation into the neurosymbolic picture.

Pros:
1. Novel integration of an ABA framework with object detection and classification via deep learning.
2. Strong motivation that pursues explainability of artificial intelligence models.
3. Promising initial results on synthetic datasets.
4. Interpretability and human-understandable reasoning via ASP representation.

Cons:
1. ILP systems are arguably very similar in intent to that of an ABA learning system, producing rules that cover positive examples and cover none of the negative ones. It would be fair to see how these two systems fare against, as ILP has already been used extensively.
A very close example to this work is https://neurosymbolic-ai-journal.com/paper/leveraging-neurosymbolic-ai-slice-discovery.
2. The related work can be greatly expanded by looking at the area of neurosymbolic visual question answering (VQA), where many models are based on ASP.
3. Experimental evaluations are limited to synthetic datasets (SHAPES-like and CLEVR-Hans3) which are easy to learn for modern neural networks, reducing real-world applicability claims. The neural network used is described as a CNN+MLP. Why not use a modern system such as YOLO?
4. Comparisons indicate lower performance than NS-CL, especially in more complex tasks (CLEVR-Hans3).

Overall, while the paper makes a valuable conceptual contribution, the experimental setup, scalability issues and lack of comparison to ILP highlight areas need significant improvement.

Minor corrections:
Section 3: aechitecture → architecture
Section 3: some notion → some notions
Section 4: First the input image → First, the input image
Section 4: using CNN → using a CNN
Section 5: Table 2 → Figure 2

**Anonymity:**

Remain anonymous

---

### Official Review · Reviewer_dEPD · 2025-04-03
**Interesting approach but paper not yet mature enough**

**Rating:** 3
**Confidence:** 4

**Review:**

Motivated by the need to improve the interpretability of image classification models, this paper proposes a novel Neuro-Argumentative Learning (NAL) architecture that combines object-centric learning via slot attention and ABA Learning. The former encodes images into facts, while the latter develops ABA frameworks that enable prediction with images. Experiments were conducted on synthetic datasets to experimentally evaluate the proposed architecture against baselines such as Neuro-Symbolic Concept Learner (NS-CL) and ResNet.

The approach seems original and interesting. However, due to several problems I report below, I believe the paper is not yet mature enough, so I opt for rejection.

- In the caption of Figure 1, the sentence "which may admit several 'extensions'" is unclear and should be better explained.
- In the "Background" section, the explanation of "Slot Attention" is difficult to follow and unclear. An intuitive explanation of its functioning would be needed, which would also help to better understand Figure 1.
- The explanation of "flat" for an ABA framework should be provided in the subsection "Assumption-Based Argumentation (ABA)".
- A description of Example 1 would help to understand it better.
- In Example 2, the fourth argument (not the third) attacks the second. The explanation that follows should be adjusted accordingly.
- What are the positive and negative examples in the subsection "Learning ABA Frameworks"? Are they images encoded into facts? Please provide an explanation.
- In the subsection "Inputs", the definition of set X is missing.
- In the subsection "Inputs", briefly explain why the neural component of the architecture disregards the labels in L.
- What is meant by "reconstructed image x^"?
- The Binary Cross Entropy (BCE) loss explanation is difficult to follow. Please try to make it clearer.
- Please briefly explain the Hungarian matching algorithm and how it was used.
- The subsection "Symbolic Component" is difficult to follow and understand. Using a running example to explain the various steps would be better.
- For ease of understanding, it would be better to have one figure for the learning architecture and another for the inference pipeline.
- In the subsection "Experiments", Table 2 should be replaced with Figure 2.
- Some considerations in the experimental evaluations are questionable. For example, the F1 score of 0.68 improves over ResNet by 7% and not 10%. Furthermore, NS-CL obtained an F1 score of 84%, which is better than that of NAL by a large amount of 16%. In addition, considerations are made about the system's execution times without providing data about them.
- There are some typos and style issues, such as wrong parentheses in citations or missing capital letters in section titles, that should be fixed.

**Anonymity:**

Remain anonymous

---

### Official Review · Reviewer_zTC9 · 2025-04-05
**Interesting proposals that add argumentation to image understanding, reasonable validation but alas only on artificial images.**

**Rating:** 7
**Confidence:** 4

**Review:**

The authors present an interesting combination of neural and symbolic methods, focusing on image understanding, and looking for ways to process features extracted by neurally inspired methods with symbolic argumentation. The theme is very relevant to the meeting. The paper is well written for the most part (a few suggestion for improvement can be found below). The contribution is valuable but a bit convoluted at times, as it is not really clear that the proposals are in fact useful at their current stage --- the empirical validation is interesting but rather limited by the fact that it is restricted to artificial images (that seem to be really artificial).

Concerning the presentation, a few suggestions:
- The review of Assumption-Based Argumentation in Section 3 is well done, with a clear explanation as to why this is a reasonable idea, and some intuition about it; on the other hand, the review of Slot Attention is very concise and indeed quite poor as it does not really provide any motivating rationale for the techniques. Why does it make sense to use such an arbitrary set of operations on data?
- Concerning Example 1, what does it really mean? Is it simply a number of labels attached to images, and a rule \rho_4 that is quite hard to grasp --- what does this rule convey? What is, after, the desired goal for this program? Please explain the example in more engaging terms.
- Small point: missing comma in Expression (2).
- What is the "Hungarian matching algorithm" actually doing at the end of Page 5?
- The symbolic component explained in Page 6 is interesting, but it is explained in such a way that the reader will hardly be able to reproduce results. Additional details would be valuable.
- As written above,  the fact that only artificial images are used does hurt the analysis. And the programs shown in Figures 4, 5, 6 seem rather simplistic; is there a need for so much machinery to impose such simple rules? Or are things more complicated to the point that the power of logic programming is really used? Please comment on this.
-

**Anonymity:**

Remain anonymous